# Reproductive Outcomes and Fertility Preservation Strategies in Women with Malignant Ovarian Germ Cell Tumors after Fertility Sparing Surgery

**DOI:** 10.3390/biomedicines8120554

**Published:** 2020-11-30

**Authors:** Francesca Maria Vasta, Miriam Dellino, Alice Bergamini, Giulio Gargano, Angelo Paradiso, Vera Loizzi, Luca Bocciolone, Erica Silvestris, Micaela Petrone, Gennaro Cormio, Giorgia Mangili

**Affiliations:** 1IRCCS San Raffaele, Department of Obstetrics and Gynecology, 20132 Milan, Italy; vasta.francescamaria@hsr.it (F.M.V.); bergamini.alice@hsr.it (A.B.); bocciolone.luca@hsr.it (L.B.); petrone.micaela@hsr.it (M.P.); 2IRCCS Istituto Tumori “Giovanni Paolo II” Gynecologic Oncology Unit, 70124 Bari, Italy; miriamdellino@hotmail.it (M.D.); giuliogar@libero.it (G.G.); ericasilvestris85@gmail.com (E.S.); 3Unit of Obstetrics and Gynaecology, Department of Biomedical Sciences and Human Oncology, University of Bari “Aldo Moro” 70124 Bari, Italy; a.paradiso@oncologico.bari.it; 4IRCCS Istituto Tumori “Giovanni Paolo II”, Scientif Director, 70124 Bari, Italy; vera.loizzi@uniba.it (V.L.); gennaro.cormio@uniba.it (G.C.)

**Keywords:** malignant germ cell ovarian tumors, fertility sparing surgery, fertility preservation

## Abstract

Malignant ovarian germ cell tumors are rare tumors that mainly affect patients of reproductive age. The aim of this study was to investigate the reproductive outcomes and fertility preservation strategies in malignant ovarian germ cell tumors after fertility-sparing surgery. Data in literature support that fertility-sparing surgery is associated with an excellent oncological outcome not only in early stages malignant ovarian germ cell tumors but also in advanced stages. Moreover, the possibility of performing conservative treatment should be considered even in case of relapse or advanced disease, given the high chemosensitivity. Indeed, available data have shown that menstrual function is maintained after platinum-based regimens in over 85–95% of patients with malignant ovarian germ cell tumors and rate of premature menopause reported in literature ranges between 3% and 7.4%, while premature ovarian failure rates are between 3.4% and 5%. Moreover, reproductive outcomes are about 80% with no increase in the risk of teratogenicity compared to general population. Therefore, conservative surgery for malignant ovarian germ cell tumors currently may represent a therapeutic option in patients who wish to preserve fertility but must be available for extended follow-up and after subscribing to informed consent.

## 1. Introduction

Malignant ovarian germ cell tumors (MOGCTs) are rare ovarian tumors accounting for approximately 5% of all ovarian malignancies [1] with an estimated incidence of 0.5 per 100,000 women [2]. They comprise a heterogeneous group of histopathological variants originating from primordial germ cells in the embryonic gonad, including dysgerminoma (DG), embryonal carcinoma (EC), immature teratoma (IT), yolk sac tumor (YST), non-gestational choriocarcinoma (CC) and mixed malignant ovarian germ cell tumor cell tumor (mMOGCT). The highest rates are encountered in women between 15 and 30 years of age [3], as these tumors represent the 80% of the preadolescent malignant ovarian tumors [4]. Rapid growth, large tumor size and high chemosensitivity are typical of these tumors. Most MOGCTs are diagnosed at an early stage with an excellent prognosis and a 5-year survival rate above 90%. Actually, a recent population-based study reports that almost 40% of patients have advanced stage disease at first diagnosis [5]. Nevertheless, the extremely good prognosis was due to the introduction in the 1980s of platinum based chemotherapy, with BEP (bleomycin, etoposide, and cisplatin) regimen leading to a survival rate close to 100% and 75% in early and advanced stages, respectively [6]. Due to their low incidence, several aspects of treatment still have to be clarified. Considering that MOGCTS arise predominantly in young women, one of the main objectives is to minimize the acute and long-term toxicities and long-term effects of cancer treatment [7] while preserving fertility. According to the most recent National Comprehensive Cancer Network (NCCN) clinical practice guidelines, the standard treatment for MOGCTs is fertility sparing surgery (FSS) regardless of cancer stage when fertility preservation is desired, in association with comprehensive surgical staging (CSS). This comprises peritoneal biopsies, omentectomy or omental biopsy, and peritoneal washing with or without retroperitoneal lymph node status assessment. CSS is mandatory to determine the extent of disease, to provide prognostic information and to tailor post-operative management [8,9]. Currently, the standard treatment after surgery is adjuvant platinum-based chemotherapy. The need for adjuvant treatment in early stage disease according to the most recent guidelines is still controversial. The European Society for Medical Oncology (ESMO) clinical practice guidelines of 2018 support surveillance instead of adjuvant chemotherapy in properly staged patients with stage IB-IC D and stage IA G2-G3 IT with negative postoperative tumor markers and to reserve active treatment in case of recurrent disease [4]. However, in the NCCN guidelines version of 2019, only stage IA-IB-IC D and stage I G1 IT can be managed with observation after surgery, while for stage IG2-G3 IT adjuvant chemotherapy is recommended [10].

## 2. Fertility Sparing Surgery (FSS) in MOGCTs

The first therapeutic approach to MOGCTs is usually surgical and it consists of FFS with CSS [4,10]. FSS is considered ontologically safe and allows to preserve one ovary and the uterus. This surgical approach should be offered to young patients with childbearing potential and to prevent early menopause. Ultraconservative surgery (cystectomy) has also been evaluated among FSS procedures in these patients setting. Beiner et al. reported eight cases of stage I immature teratoma treated by cystectomy followed by chemotherapy in five cases, documenting no recurrences after a median follow-up of nearly 5 years [11]. Despite this, according to current guidelines, unilateral salpingo-oophorectomy (USO) is the recommended procedure in case of unilateral involvement. For years, contralateral ovarian biopsy has been routinely performed in case of FSS; given some recent evidence, this procedure should be avoided when the ovary appears macroscopically free of injury, due to an increased risk of peritoneal adhesions, a potential cause of mechanical sterility [12,13]. For those patients who decide not to pursue fertility preservation, other quality of life issues should be taken into consideration to tailor the best surgical approach. The direct impact on the physical and psychosocial aspects of the sexual function that a demolitive surgery would entail should not be underestimated [14]. Furthermore, early surgical menopause is associated with more severe short-term effects than spontaneous menopause, such as moderate to severe vasomotor symptoms, more severe sleep disturbance, and overall impaired quality of life [15]. The long-term consequences of premature or early menopause include adverse effects on cognition, mood, cardiovascular, bone, and sexual health, as well as an increased risk of early mortality [16]. Because MOGCTs are most commonly associated with several types of gonadal dysgenesis, when these abnormalities are discovered, FSS cannot be performed [17]. Due to the greater risk for developing neoplasia, these patients should undergo bilateral oophorectomy, while the uterus can be preserved for a possible embryo transfer [18]. Several retrospective studies have shown how FSS instead of a demolitive approach does not impair oncologic outcomes in early stage MOGCTs. Given the rarity of this disease, prospective data are not available. Principal published series are summarized in Table 1.

One of the first studies was conducted in 1977 by Kurman and Norris. In a cohort of 182 patients with early stage MOGCTs undergoing conservative surgery, they did not observe a worsened prognosis [19]. Subsequent studies have supported these data, showing that FSS is feasible and safe not only in early stages, but also in advanced stage disease. Due to their high chemosensitivity and the rare massive bilateral ovarian involvement, all stages MOGCTs can be treated with FSS with excellent results on both survival and reproductive outcome. In 1995, Peccatori et al. retrospectively evaluated the surgical management of 129 patients, reporting satisfactory results in term of recurrence and cure in 68 patients with stage I and 40 patients with stage II–IV disease who underwent FSS, with an overall survival of 96% [20]. A large population-based analysis from Surveillance, Epidemiology, and End Results (SEER) database, including data of 535 patients with all disease stages, from 1988 to 2001, was conducted by Chan et al. They have reported a progressive increase in the use of FSS over the years, with a satisfactory survival rate both in early and advanced stages MOGCTs [8]. In a recent study, Park et al. evaluated the prognosis of 171 patients (79 women at stage I and 46 women at stages II–IV) who underwent FSS for MOGCTs with a median follow-up time of 86 months, confirming that FSS did not have a negative effect on OS or PFS not only in early stages but also in advanced stages [21]. Bilateral localization of MOGCTs is extremely rare with a prevalence of 4.3% to 6.9% of cases, even though in case of dysgerminomas bilaterally occurs in up to 15% of cases [22,23,24]. Notably, bilateral involvement is not always of malignant nature, considering that 5–10% of patients with MOGCTs may present a benign cystic teratoma on the contralateral ovary [25]. In case of macroscopic bilateral ovarian involvement, unilateral salpingo-oophorectomy with contralateral cystectomy is recommended, in order to maintain fertility and ovarian function [2]. However, if both ovaries are almost replaced by neoplastic tissue, the risk of leaving residual disease is very high and it has to be correlated to the histological type. Data in literature support that, especially when the bilateral tumor is a dysgerminoma, due to the high chemosensitivity, patients maintain a satisfactory prognosis. In a MITO-9 retrospective study, Sigismondi et al. reviewed 8 of 145 patients (5.5 %) affected by bilateral tumors and four of them received FSS [26]. In total, 50% of these patients, of 13 and 26 years old, respectively, were affected by mMOGCTs with both ovaries massively replaced by neoplastic tumor, precluding any chance to visualize and save any intact ovarian parenchyma. In both cases, only a biopsy of the smaller ovary with suspected macroscopic characteristics for dysgerminoma histology was performed, leaving almost all the ovarian neoplastic tissue in place. At a median follow-up of 54 months, all these patients were still alive with no evidence of disease, showing that FSS in case of bilateral MOGCTs does not compromise the oncologic outcomes [26]. Vicus et al. reported the case of a 17-year-old patient with bilateral dysgerminoma stage IB in which residual disease was intentionally left in the ovary to preserve fertility, after completing the treatment with platinum-based chemotherapy the patient had remained relapse-free for 5 years [27]. In case of advanced disease and/or poor performance status, when primary surgery is not feasible, multiagent platinum based neoadjuvant chemotherapy (NACT) is a possible option. Given the high chemosensitivity of these tumors, FSS still remains feasible even in patients with advance disease [28]. Lu et al. reported that NACT can guarantee a better optimal cytoreduction rate and less peri-operative morbidities in patients with unresectable tumors and poor general condition [29]. Talukdar et al. have recently reported that 21 (91.3%) of 23 patients treated with 4 cycles of neoadjuvant BEP followed by FSS were alive and disease-free at a median follow-up of 74 months [30]. A total of 18 of 21 patients have resumed menstruation and 10 eligible patients have delivered 13 full-term healthy babies, also reporting good results about fertility status maintenance [30]. In those cases where it is impossible to preserve fertility by USO, FSS can be performed by bilateral salpingo-oophorectomy (BSO) preserving only the uterus for future embryo transfer. In 2017 Nasioudis et al. conducted a big retrospective study including a total of 526 patients with advanced stages (II-IV) to evaluate the impact of sparing the uterus on survival. With a rate of uterine preservation of 79.8%, no impact on survival was found [31].

## 3. Relapse after Conservative Treatment of MOCGTs: Risk Factors and Management

Several studies in literature have attempted to identify risk factors for recurrence, with controversial results. Histological type, incomplete peritoneal surgical staging, residual tumor, age, and stage at diagnosis are considered possible risk factors for recurrence, while surgical approach does not seem to affect the relapse rate according to available data [32]. Unfortunately, given the rarity of this disease, no prospective data are available. Several studies suggest that YST is as an independent poor prognostic factor in MOGCTs [33]. Among the largest series available, Chan et al. have analyzed the outcome after FSS of 760 MOGCTs patients included in the SEER database, reporting that older age, YST histology, and advanced stage are significant independent prognostic factors for poorer survival [8]. Typically, most of the recurrences occur within 2 years from initial diagnosis. According to a retrospective study by Mangili et al. including 123 patients with MOGCTs stage I–IV, 81 (65.9%) treated with FSS, recurrence rate was 17.8%. More than half of the recurrences occurred within 24 months from diagnosis [34]. CSS is crucial in early stages to select those patients who really need to receive adjuvant chemotherapy, but also to considerably decrease the recurrence rate [35,36,37]. Data from 144 patients with stage I disease, retrospectively collected among MITO centers showed that the recurrence rate was higher in the incomplete staged compared to the surgically staged group (26% vs. 8.6%), with incomplete peritoneal staging being an independent risk factor for recurrence [38,39]. The management of patients with recurrent and persistent MOGCTs is still a challenge due to the rarity of these conditions, as no standardized protocol for salvage therapy is currently available [40]. In case of a suspected relapse, surgery is mandatory to define whether active (=malignant) disease or benign tumor is present. In fact, benign germ cell tumors (e.g., mature teratoma) are frequently found in MOGCTs patients, and do not need further treatment after surgical removal. The possibility of performing conservative treatment should be considered even in case of relapse or advanced disease, given the high chemosensitivity [40]. Treatment of relapse should be tailored to each patient, according to previous therapies received. Patients relapsing after surveillance could be addressed to platinum based multiagent chemotherapy, with BEP regimen the most commonly used. Those patients relapsing after platinum chemotherapy represent a treatment challenge. TIP (paclitaxel, ifosfamide, and cisplatin) is recommended as second-line therapy, other therapeutic options include VIP (vinblastine, ifosfamide, and cisplatin) and gem- TIP (gemcitabine, paclitaxel, ifosfamide, and cisplatin) [41]. High-dose chemotherapy (HDC) with peripheral blood progenitor cell (PBPC) support is considered an option for salvage treatment in patients with platin resistance [42].

## 4. Reproductive Outcomes after MOGCTs Treatment

Only a few studies have been conducted on the reproductive outcomes after FSS in early and advanced stage MOGCTs and evidence concerning fertility potential remain unclear [42]. This is because MOGCTs are rare tumors and only about 30 years have passed since the dramatic improvement in prognosis with BEP therapy. However, among significant BEP-related toxicities, fertility impairment due to follicle destruction, ovarian stromal fibrosis and reduction of primordial follicles is one of the most relevant [43,44,45]. These effects seem to be strictly correlated to the type of drugs used, schedule, total dose, and treatment duration [46]. In an attempt to diminish toxicity, JEB (carboplatin, etoposide, and bleomycin) has replaced BEP in children [47], given that among chemotherapeutic agents, cisplatin has the highest risk of causing ovarian failure [48]. Available data have shown that menstrual function is maintained after platinum-based regimens in over 85–95% of patients with MOGCTs [49,50]. Brewer et al. reported that 71% of patients who underwent FSS and BEP chemotherapy maintained their normal menstrual function during and after treatment, the remaining patients regained normal menses within 6 months of completing chemotherapy [51]. However, resumption of cyclic menses after oncologic treatment does not necessarily imply normal fertility. The rate of premature menopause reported in literature ranges between 3% and 7.4%, while premature ovarian failure rates are between 3.4% and 5% [34,52,53]. Data in literature regarding reproductive outcomes are reassuring. Results from available studies are summarized in Table 2.

Zanetta et al. described the obstetric outcome of 128 potentially fertile conservatively treated patients with MOGCT. They reported that 27.4% of patients treated with adjuvant chemotherapy and 21.8 % of patients treated only with FSS attempted to conceive, with a pregnancy rate of 80% and 100%, respectively [12]. Yang et al., in 2018, published data of their study about 148 patients with MOGCTs treated with FSS, of whom 129 also received adjuvant chemotherapy [54]. They reported that among the patients who completed chemotherapy none developed amenorrhea and that almost 80% of those who desired pregnancy conceived with a total of 35 live births without congenital anomalies [55]. Similar results have been achieved by Park et al. with a pregnancy rate of 75% and a live birth rate of 65% [56]. On the other hand, Ertas et al. evaluated the reproductive outcomes in 39 patients treated for MOGCTs with FSS they found no significant differences between patients who received adjuvant chemotherapy and those who received observation alone, with fertility rates of 77% and 62.5%, respectively [57]. Nevertheless, this study had limitations due to the small patient sample. With the same intent to clarify the reproductive outcomes in MOGCTs survivors Tamauchi et al. in 2018 identified a total of 110 MOGCT who received FSS with a median follow-up of 10.4 years. They reported that >90% of the 45 patients who desired to bear children became pregnant [53]. They also showed that the relatively long period between MOGCTs treatment and achieving pregnancy is due to the young age at diagnosis and the increasing age at first marriage or childbirth. To note, the trend toward higher age at first pregnancy is associated with an increased risk of infertility [58]. Further evidence is therefore needed to evaluate the impact of antiproliferative treatments on gonads in relation to the physiological reduction of the ovarian reserve with increasing age. In 2014, Solheim et al. conducted the first study to assess the relation between post-treatment fertility and cumulative dose of cisplatin in MOGCTs survivors. They reported that more than three cycles of cisplatin-based chemotherapy were associated with higher risk of premature menopause, higher infertility rate and longer median conception intervals. However, a satisfactory conception rate was reported in this patient population, as 87.1% of patients who had attempt post-treatment pregnancy become pregnant at least once [59].

According to literature available on the possible dangerous effects of cancer treatments on pregnancy, no evidence of an increased risk of genetic or other defects in new-born from MOGCTs survivors exists [60]. Data are currently scant, and further and long-term studies should be conducted to confirm this. However, available evidence is reassuring, reporting an infertility rate of about 20%. Therefore, a young woman with MOGCTs at any stage must be reassured about the promising oncologic and fertility outcomes after treatment.

## 5. Fertility Preservation Strategies

Fertility counselling should become an integral part of the clinical management of women with MOCGTs (Figure 1).

Unfortunately, this is not always feasible before surgery, because diagnosis in not yet known. Moreover, in most cases, patients present with an acute abdomen resulting from ovarian torsion, hemorrhage or tumor rupture that requires emergency intervention; therefore, diagnosis is incidentally made [61,62].

The first fertility preservation strategy to be pursued in MOGCTS is FSS, also in case of bilateral involvement. Reproductive assessment should be also performed after surgery, and should be related to the postoperative strategy (surveillance or adjuvant chemotherapy). Several studies have shown that reproductive capacity is not particularly affected after treatment; however, in some cases it might negatively impact on ovarian reserve affecting the duration of the fertile period [63,64]. Ovarian reserve should be evaluated at least 6 months after treatment following the rebalancing of the ovarian function [65,66]. Anti-Mullerian Hormone (AMH) should be routinely assessed, with the aim to promptly identify an impair of the reproductive function and to propose fertility preserving options [67]. Several randomized clinical trials have evaluated the effectiveness of LHRH (Luteinizing Hormone Releasing Hormone) analogues in the prevention of chemotherapy-induced ovarian failure [68,69,70]. However, since chemotherapy regimens used for treating MOGCTs do not show particular toxicity, the association with LHRH analogues might not change the effects on fertility [37,71] Moreover, as literature reported, LHRH analogues increase non-significantly the chances of menstrual recovery [72]. Several fertility preservation techniques can be offered to MOGCTs patients. They include oocytes cryopreservation following controlled ovarian hyper-stimulation (COH) The standardized method is oocyte cryopreservation, which could be proposed to patient with childbearing potential and in case of unilateral MOCGTs. After surgery, if surveillance is an option, in addition to oncological follow-up, reproductive-endocrine evaluation must be performed in these patients. If chemotherapy is needed after surgery, it should be delayed by at least 10 to 12 days to allow time for oocyte maturation [72]. Whenever possible, this is of significant importance in advanced stages, as a number greater than three cycles is usually expected, making these patients more exposed to chemotherapy-related damage. In patients who have not performed fertility preservation before chemotherapy, COH should be performed after 6-12 months after adjuvant treatment [73]. It is also important to emphasize that performing ovarian stimulation after chemotherapy leads to a reduced ovarian response, as ovaries can already be damaged. Ovarian tissue cryopreservation represents another fertility preservation option. As no ovarian stimulation is required, this is the only technique suitable for prepubertal patients. Unfortunately, the transplantation of the cryopreserved ovarian tissue is not an option for women with MOCGTs, because of the risk of spreading cancer to the contralateral ovary [73]. Oocytes cryopreservation after in vitro maturation (IVM) of oocytes collected from tissue without pharmacological stimulation is still experimental. Nevertheless, ovarian follicles can be isolated from the tissue and grown in vitro to obtain mature eggs, which can then be fertilized and eventually transferred to the uterine cavity [74]. In women who cannot receive FSS without sparing of the uterus, surrogate pregnancy should be also considered, but the application of this technique in clinical practice is still limited due to ethical and legal issues in different countries [75].

## 6. Conclusions

MOGCT is a rare disease that typically present in the teenage years. On the other hand, MOCGT in pregnancy are rare and better maternal and fetal outcomes are the results of a multi-disciplinary comprehensive approach [76]. The tumor is always unilateral and chemo-sensitive. As such, FSS is considered the procedure of choice in young women with tumor confined to a single ovary. In advanced disease, the role of aggressive surgery is not well-established and removal of both ovaries does not guarantee improvement in outcome. The gold standard for adjuvant therapy is the combination which included bleomycin, etoposide, and cisplatin. Several studies evaluating ovarian and reproductive function after conservative surgery and chemotherapy for MOGCTs have consistently demonstrated excellent prognosis. In addition, the return of normal menstrual function with normal fertility rates with no increase in the risk of teratogenicity has also been evaluated.

## Figures and Tables

**Figure 1 biomedicines-08-00554-f001:**
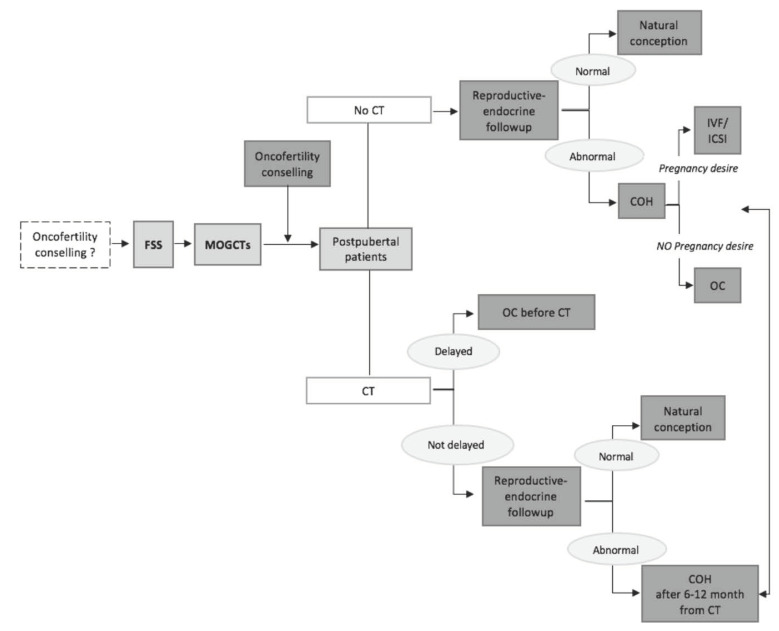
The flow-chart of fertility preservation strategies in patients with MOGCTs. Fertility sparing surgery (FSS), Malignant ovarian germ cell tumors (MOGCTs), Controlled ovarian hyper-stimulation (COH), CT (chemotherapy), Oocyte cryopreservation (OC), In Vitro Fertilization (IVF), Intra Cytoplasmic Sperm Injection (ICSI).

**Table 1 biomedicines-08-00554-t001:** Review of studies comparing oncological outcomes between conservative and demolitive surgery in MOGCTs.

			Conservative Surgery			Demolitive Surgery		
Study, Year	Tot Patient n	Median Follow up (Months)	n° Patients	N Patient for Stages	% Survival	n° Patients	% Survival	% Survival Tot (Conservative+ Demolitive)
Peccatori et al. (1995)	19	55	108	Stage I: 68 Stage II–IV: 40	96%	21		
Lee et al. (2009)	196	67	134			62		Stage I–II: 98.7% Stage III–IV: 92.7%
Perrin et al. (1999)	45	58.7		Stage I: 36 Stage II–IV: 9	Stage I: 97% Stage II–IV: 88%			
Low et al. (2000)	91	52.1	74	Stage I: 56 Stage II–IV: 18	Stage I: 98.2% Stage II–IV: 94.4%	17		
Chan et al. (2008)	535	150	313		98 %	122	96%	Stage I–II: 97.6% Stage III–IV: 85.5%
Mangili et al. (2011)	123	60.9	92	Stage I: 71 Stage II–IV: 21	Stage I: 97.1% Stage II–IV: 71.4%	31	81%	
Zanetta et al. (2001)	169	67	138	Stage I: 88 Stage II–IV: 50	98%	31	87%	95%
Tangjitgamol et al. (2010)	124	63.6	89		93%	35	91%	
B. Yang et al. (2018)	122	60	122		96.7%	-		
Park et al. (2017)	199	86	171	Stage I: 125 Stage II–IV: 46	Stage I: 99% Stage II–IV: 91%	28	-	
Z. Yang et al. (2014)	106	56.5	59	Stage I–II: 41 Stage III–IV:18	84.7%	45		
TOTAL	1839							

**Table 2 biomedicines-08-00554-t002:** Reproductive outcomes after conservative treatment in MOGCTs.

Study (year)	n° Patients Undergoing Conservative Surgery	n° Patients Evaluable after FSS	n° Patients Tried to Conceive	n° Patients Getting Pregnant
Low et al. (2000)	74	72	20	19/20 (95%)
Mangili et al. (2011)	92	84	15	12/15 (80%)
Weinberg et al. (2011)	22	14	10	8/10 (80%)
Zanetta et al. (2001)	138	128	32	28/32 (87.5%)
Tangir et al. (2003)	64	64	38	29/38 (76%)
Zanagnolo et al. (2005)	33	19	9	6/9 (66.6%)
Park et al. (2017)	171	124	20	15/20 (75%)
Z. Yang et al. (2018)	59	59	39	31/39 (79.5%)
Ertas et al. (2014)	41	35	21	15/21 (71.4%)
Solheim et al. (2015)	61	61	39	34/39 (87.2%)
Tamauchi et al. (2018)	110	105	45	42/45 (93.3%)
Vicus et al. (2010)	50	50	15	12/15 (80%)
Bonazzi et al. (1994)	31	31	6	5/6 (83.3%)

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
