# Peer review of "Reproductive Outcomes and Fertility Preservation Strategies in Women with Malignant Ovarian Germ Cell Tumors after Fertility Sparing Surgery"

_biomedicines, 2020, doi:10.3390/biomedicines8120554_

Round 1
Reviewer 1 Report
The review is very interesting and very good structurate. I suggest few minor language and editing corrections.
Point 1. The abstract needs to be modified because its not easy to follow and avoided abbreviators.
Point 2. In this sentence the author's should change into account with in consideration: For those patients who decide not to pursue fertility preservation, other quality of life issues should be taken into account to tailor the best surgical approach.
Author Response
REQUEST FOR REVISION (16/11/2020) -REVISION (19/11/2020)
Answers to Reviewers
Point 1. The abstract needs to be modified because it’s not easy to follow and avoided abbreviators.
Ok, done.
Point 2. In this sentence the authors should change into account with in consideration: For those patients who decide not to pursue fertility preservation, other quality of life issues should be taken into account to tailor the best surgical approach.
Ok, done.
Thank you for revision.
Reviewer 2 Report
“Reproductive outcomes and fertility preservation strategies in women with malignant ovarian germ cell tumors after fertility sparing surgery” by Francesca Maria Vasta and Colleagues).
A professional editor is recommended to proof the grammar and syntax of the manuscript.
Throughout the manuscript, the authors sometimes cite two references, such as [3][4] or [3,4]. Please follow the second form.
Abstract section
The first sentence in the abstract means to be a running title? Please clarify.
Please define FSS and MOGCTs in their first appearance in the abstract and then use the abbreviation.
Please correct: “7.4%” not “7,4%”.
Keywords section
Since the authors defined MOGCTs as malignant ovarian germ cell tumors, this designation should be maintained in the keywords and not malignant germ cell ovarian tumors. This is just advice.
Introduction section
In the definition of the abbreviation mMOGCTs please correct for mixed malignant ovarian instead of mixed ovarian malignant.
Table 1: Please use lowercase to write “comparing”. Some data in this table are not clear. For example, the number of patients and the % of survival are in relation to what surgery, conservative or demolitive? The authors need to reformulate this table in order to make it more reader friendly.
Page 4, line 11: Please define OS and PFS, as it is their first appearance.
Page 4, line 21: Please define MITO-9.
In relation to the study of Sigismondi et al [27], how old were these two patients and what is their fertile status?
When primary surgery is not feasible, multiagent platinum based neoadjuvant chemotherapy is a possible option, but there are results already regarding this procedure. Since this will delay surgery, or prevent surgery in case of unresectable tumors will the “good” prognosis be maintained? And what about fertility status maintenance? Please clarify.
Page 4, line 39: Please define BSO.
Page 5, line 4: Please correct “risk factors” instead of “risk factor”.
Table 2: It would be important to define the median age of the patients undergoing conservative surgery and the median age of the patients that tried to conceive. These pregnancies are all natural or there is any that resorted the assisted reproductive medicine? What about the reasons for not trying to conceive, is it possible to disclaim some? Please clarify these points.
Page 7, line 9: Please use MOGCT instead of the designation malignant ovarian germ cell tumor.
In the flowchart of figure 1, why were not the prepubertal girls included? Please insert in the legend of this figure the definition of all abbreviations used.
Page 9, line 1: Please define LHRHs.
These two references should be included in the manuscript and analysed:
Choi MC, Chung YS, Lee JW, Kwon BS, Park BK, Kim SI, Shim SH, Lee KB, Seong SJ, Lee SJ, Lee SH, Yoo HJ, Song T, Kim MK, Baek MH, Kang S, Kim YM. Feasibility and efficacy of gonadotropin-releasing hormone agonists for the prevention of chemotherapy-induced ovarian insufficiency in patients with malignant ovarian germ cell tumours (KGOG 3048R). Eur J Cancer. 2020;133:56-65.
Luh LCPN, Mahendra INB, Suwiyoga K, Budiana I, Mayura IM, Wiradnyana AP, Putra IWA, Negara IKS, Ariawati K, Dewi ISM, Susraini AN, Sriwidyani NP, Wirawan W, Vikananda IA. Management Comprehensive Multidisciplinary of Malignant Ovarian Germ Cell Tumors and Feto - Maternal Outcome: A Case Series Report and Literature Review. Open Access Maced J Med Sci. 2019; 7:1174-1179.
Authors contributions
The authors already acknowledged the funding and specified the funding, please stick to that. Funding support is not an author contribution.
References section
Please follow the journal guidelines:
- Abbreviate the journal’s name of reference 14.
- Leave a space between the journal’s name and the year of publication.
- The authors have to endpoints in reference 37.
- Correct the journal’s name of reference 52.
- Correct reference 54. Volume is not correctly presented.
- Remove the month of publication from reference 57.
- Separate the year of publication from the volume by “;” and not “,”.
- Do not abbreviate the last page of the manuscripts.
- Remove issues of the publications.
- Put the year of the references in bold.
Therefore, it is this reviewer’s recommendation to review the manuscript taking into account the concerns raised before its acceptance for publication in Biomedicines.
Author Response
REQUEST FOR REVISION (16/11/2020) -REVISION (19/11/2020)
Answers to Reviewers
REQUEST FOR REVISION (16/11/2020) -REVISION (19/11/2020)
Point 1. The abstract needs to be modified because it’s not easy to follow and avoided abbreviators.
Ok, done.
Point 2. In this sentence the authors should change into account with in consideration: For those patients who decide not to pursue fertility preservation, other quality of life issues should be taken into account to tailor the best surgical approach.
Ok, done.
Throughout the manuscript, the authors sometimes cite two references, such as [3][4] or [3,4]. Please follow the second form.
Where?
Abstract section
The first sentence in the abstract means to be a running title? Please clarify.
Ok, done.
Please define FSS and MOGCTs in their first appearance in the abstract and then use the abbreviation.
Ok, done.
Please correct: “7.4%” not “7,4%”.
Ok, done,
Keywords section
Since the authors defined MOGCTs as malignant ovarian germ cell tumors, this designation should be maintained in the keywords and not malignant germ cell ovarian tumors. This is just advice.
Ok, done.
Introduction section
In the definition of the abbreviation mMOGCTs please correct for mixed malignant ovarian instead of mixed ovarian malignant.
Ok, done.
Table 1: Please use lowercase to write “comparing”. Some data in this table are not clear. For example, the number of patients and the % of survival are in relation to what surgery, conservative or demolitive? The authors need to reformulate this table in order to make it more reader friendly. Ok, done.
Page 4, line 11: Please define OS and PFS, as it is their first appearance.
Ok, done.
Page 4, line 21: Please define MITO-9.
Ok, done.
In relation to the study of Sigismondi et al [27], how old were these two patients and what is their fertile status? The patients with bilateral tumors who received FSS were 13 and 26 years old, they both resumed normal menstruations.
When primary surgery is not feasible, multiagent platinum based neoadjuvant chemotherapy is a possible option, but there are results already regarding this procedure. Since this will delay surgery, or prevent surgery in case of unresectable tumors will the “good” prognosis be maintained? And what about fertility status maintenance? Please clarify. Ok, done.
Page 4, line 39: Please define BSO.
Ok, done.
Page 5, line 4: Please correct “risk factors” instead of “risk factor”.
Ok, done.
Table 2: It would be important to define the median age of the patients undergoing conservative surgery and the median age of the patients that tried to conceive. These pregnancies are all natural or there is any that resorted the assisted reproductive medicine? What about the reasons for not trying to conceive, is it possible to disclaim some? Please clarify these points.
Unfortunately, this data is not recoverable from all articles correctly, so we prefer not to include this field in the table.
Page 7, line 9: Please use MOGCT instead of the designation malignant ovarian germ cell tumor.
In the flowchart of figure 1, why were not the prepubertal girls included? Please insert in the legend of this figure the definition of all abbreviations used.
In prepubertal patients, cryopreservation of ovarian tissue, although still considered experimental, is the only method available; it does not require ovarian stimulation or sexual maturity and therefore is currently the reference technique in these patients. Unfortunately, the transplantation of the cryopreserved ovarian tissue is not an option for women with MOCGTs, because of the risk of spreading cancer to the contralateral ovary, for this reason they have not been included in the flow-chart.
Page 9, line 1: Please define LHRHs.
Ok, done.
These two references should be included in the manuscript and analysed:
Choi MC, Chung YS, Lee JW, Kwon BS, Park BK, Kim SI, Shim SH, Lee KB, Seong SJ, Lee SJ, Lee SH, Yoo HJ, Song T, Kim MK, Baek MH, Kang S, Kim YM. Feasibility and efficacy of gonadotropin-releasing hormone agonists for the prevention of chemotherapy-induced ovarian insufficiency in patients with malignant ovarian germ cell tumours (KGOG 3048R). Eur J Cancer. 2020; 133:56-65.
Luh LCPN, Mahendra INB, Suwiyoga K, Budiana I, Mayura IM, Wiradnyana AP, Putra IWA, Negara IKS, Ariawati K, Dewi ISM, Susraini AN, Sriwidyani NP, Wirawan W, Vikananda IA. Management Comprehensive Multidisciplinary of Malignant Ovarian Germ Cell Tumors and Feto - Maternal Outcome: A Case Series Report and Literature Review. Open Access Maced J Med Sci. 2019; 7:1174-1179.
Ok, done.
Authors contributions
The authors already acknowledged the funding and specified the funding, please stick to that. Funding support is not an author contribution.
References section
Please follow the journal guidelines:
Abbreviate the journal’s name of reference 14. Ok, done.
Leave a space between the journal’s name and the year of publication. Ok, done.
The authors have to endpoints in reference 37. Ok, done
Correct the journal’s name of reference 52. Ok, done.
Correct reference 54. Volume is not correctly presented. Ok, done.
Remove the month of publication from reference 57. Ok, done.
Separate the year of publication from the volume by “;” and not “,”. Ok, done.
Do not abbreviate the last page of the manuscripts. Ok, done.
Remove issues of the publications. Ok, done.
Put the year of the references in bold. “,”. Ok, done.
Therefore, it is this reviewer’s recommendation to review the manuscript taking into account the concerns raised before its acceptance for publication in Biomedicines.